# Flavor Formation in Dry-Cured Fish: Regulation by Microbial Communities and Endogenous Enzymes

**DOI:** 10.3390/foods12163020

**Published:** 2023-08-11

**Authors:** Jiayue Liu, Ruijie Mai, Pingru Liu, Siqi Guo, Juan Yang, Weidong Bai

**Affiliations:** 1College of Light Industry and Food Technology, Zhongkai University of Agriculture and Engineering, Guangzhou 510408, China; jiayuezan@163.com (J.L.); mairuijie@foxmail.com (R.M.); pingruliu@163.com (P.L.); ckiskitty@163.com (S.G.); whitebai2001@163.com (W.B.); 2Guangdong Provincial Key Laboratory of Lingnan Specialty Food Science and Technology, Zhongkai University of Agriculture and Engineering, Guangzhou 510408, China; 3Key Laboratory of Green Processing and Intelligent Manufacturing of Lingnan Specialty Food, Ministry of Agriculture, Beijing 430062, China; 4Academy of Contemporary Agricultural Engineering Innovations, Zhongkai University of Agriculture and Engineering, Guangzhou 510408, China

**Keywords:** dry-cured fish, microbial community, flavor formation mechanism of dried salted fish, volatile flavor substances, endogenous enzymes

## Abstract

Dried salted fish is a traditional dry-cured fish that is sprinkled with salt before the curing process. With a unique flavor as well as diverse varieties, dry-cured fish is popular among consumers worldwide. The presence of various microbial communities during the curing process leads to numerous metabolic reactions, especially lipid oxidation and protein degradation, which influence the formation of flavor substances. However, during industrial curing, the quality of dry-cured fish is difficult to control, leading to the formation of products with diverse flavors. This review describes the curing process of dried salted fish, the key microorganisms involved in the curing process of typical dried salted fish products at home and abroad, and the correlation between biological metabolism and flavor formation and the underlying mechanism. This review also investigates the prospects of dried salted fish products, proposing methods for the analysis of improved curing processes and the mechanisms of dried salted fish. Through a comprehensive understanding of this review, modern production challenges can be addressed to achieve greater control of microbial growth in the system and improved product safety. In addition to advancing our understanding of the processes by which volatile flavor compounds are formed in conventional dry-cured fish products, we expect that this work will also offer a theoretical framework for enhancing their flavor in food processing.

## 1. Introduction

Dry-cured fish is a traditional processed fish product. Dry salting involves the direct application of salt to fresh fish meat; the fish is dehydrated under the high osmotic pressure of salt, which dissolves into salt water and gradually seeps into the raw materials. Dry curing was originally used for the improved storage and preservation of fresh fish but is now popular among consumers because of the use of solid-state fermentation, which improves the flavor and nutritional value of the products. The unique flavor of salted and dried fish is mainly produced by the microbial metabolism of carbohydrates, the interaction of endogenous proteases, and the decomposition of fat. These salted and dried fish products are mostly regional. 

The basic phases of dry salted fish preparation (Figure 1) are raw material selection (the descaling and gutting of fresh fish) and washing them in freshwater. Then, the fish are treated in a salt brine for hours and are dried for days. Finally, the dry-cured fish is packaged and ready for consumption. The principal steps influencing the flavor composition are drying and curing [1]. 

For a long time, traditional dried salted and dried fish mainly used natural fermentation, which is restricted by the natural external conditions. For instance, unstable weather makes it more difficult to maintain a consistent temperature and humidity level. This process can result in poor food safety and has difficulty meeting the huge market demand. To increase the production of dried fish and to satisfy consumers’ new demands for product taste and quality, traditional salted and dried fish products must undergo industrialization development [2]. Fresh fish are rich in protein, low in fat, and nutritious. The flavor of dried salted fish products is also vital for better edibility. Research has shown that the unique flavor of dried salted fish is created by volatile flavor precursors, which are mainly free fatty acids produced by lipolysis and free amino groups produced by protein hydrolysis [3,4]; biochemical reactions involving endogenous enzymes; and a range of microorganisms. 

The main aroma substances in dried salted fish products include aldehydes, ketones, alcohols, and other small molecular flavor substances, and most of the aldehydes are derived from the oxidation and decomposition of fatty acids [5]. Lipase hydrolyzes lipids to generate flavor precursors like free fatty acids, which are then oxidized to form volatile taste molecules. The endogenous lipases in dried salted fish are mainly divided into lipohydrolase and lipoxygenase, which are closely related to the degradation and oxidation of fat and the formation of volatile flavor substances. The effects of lipid oxidation and microbial metabolism promote the formation of a unique salty flavor in dried salted fish products. The impacts of microorganisms and enzymes on flavor were summarized by combining the metabolic process with the dominating strains in the curing process of dried salted fish at home and abroad. From the perspectives of lipid hydrolysis, oxidation, and flavor formation, the relative roles of microorganisms and endogenous enzymes in lipid and flavor changes were investigated. For instance, 3-methylbutyral and phenylacetaldehyde were produced from amino acid degradation, which was related to microbial activities [6].

Herein, we review the mechanism of the formation of the main flavor substances in dried salted fish products. It is explained how microbial communities affect lipid oxidization and protein hydrolysis. By identifying the key flavor substances and dominant microorganisms in the curing process, the safety issue of salted and dried fish products can be addressed together with achieving the desired flavor and the modernization and standardization of manufactured salted and dried fish products.

## 2. Formation Pathway of Flavor Substances Derived from Lipid Oxidization or Protein Hydrolysis and Strecker Degradation in Dried Salted Fish

Protein degradation, lipid hydrolysis, and fat oxidation are crucial to the formation of the unique flavor of salted and dried fish, and endogenous and microbial enzymes play an important role [7,8]. The relative roles of microbial flora and endogenous enzymes in lipid and protein degradation and flavor formation in salted and dried fish products have been studied. Compared with microbial flora, fish lipase plays a major role in lipid hydrolysis, whereas microorganisms play a dominant role in lipid oxidation and flavor formation [9,10]. Microorganisms play a major role in protein degradation and amino acid metabolism, which is a key step in the formation of protein-derived flavor substances. The typical components detected in salted and dried fish can be divided into nine categories: aldehydes, ketones, alcohols, esters, alkanes, alkenes, aromatic compounds, amines, and other volatile substances [11].

Common flavor substances in salted and dried fish include aldehydes (e.g., hexyl aldehyde, nonal, octyl aldehyde, 3-methyl-butyral, and phenylacetaldehyde), ketones (e.g., 6-methyl-5-heptene-2-ketone; 2, 3-heptene-2-ketone; 3-heptene-2-ketone; 2-heptene-ketone; 2-nonone; 3-octyl ketone; 1-octene-3-ol; hexol; and 1-hexanol), esters (e.g., 2-methyl-butyl butyrate, methyl phthalate, ethyl acetate, and 3-methyl-1-butanol propionate), and olefins (e.g., terpenes). Common key flavor substances such as aldehydes usually have a low sensory threshold and a relatively high content in volatile flavor substances and mainly affect the flavor of dried salted fish (Figure 2) [12]. One study found that the contents of the major volatile compounds were aldehydes in four salted dried fishes, karafuto shishamo (*Spirinchus lanceolatus*), tigertooth croaker (*Otolithes rube*), little yellow croaker (*Pseudosciaena polyactis*), and hairtail (*Trichiurus haumela*), which accounted for 49.99% [13]. Flavor compounds are generated by the synergistic action of endogenous enzymes and microorganism-derived enzymes producing primary metabolites, which are then synthesized through secondary metabolism.

### 2.1. Aldehydes

Aldehydes contribute significantly to food flavor because of their low thresholds. Aldehydes can be divided into those derived from lipid oxidization or amino acid degradation [14]. Aldehydes are produced through the oxidation of linolenic acid or linoleic acid and other lipid compounds. For example, linoleic acid generates 13s-hydroperoxy compounds through lipoxygenase and then generates hexal, which has a grass flavor, through dehydrogenase activity [15]. Linolenic acid is processed by lipoxygenase to produce the intermediate product 10S-hydroperoxide, which is then converted by dehydrogenase to the rancid-tasting nonyl aldehyde [16]. Linolenic acid is synthesized by lipoxygenase as an intermediate 11S-hydroperoxide, which is then dehydrogenated into a grassy-scented octyl molecule by dehydrogenase. Aldehydes are also derived from protein sources. Leucine is degraded by leucine aminotransferase and branched amino acid aminotransferase to generate the intermediate product α-ketoisocaproic acid. This intermediate product is then metabolized by α-ketoacid decarboxylase to generate 3-methyl-butyraldehyde, which has an unpleasant odor [17]. *Bacillus* species have been found to be related to leucine aminotransferase activity, while the production of α-ketoisocaproic acid is related to *Enterobacter*, *Lactobacillus*, *Megalococcus*, and *Staphylococcus* species. Phenylalanine is converted to the intermediate product phenylpyruvic acid by the enzymes phenylpyruvic transaminase and aromatic amino acid transaminase, which is then converted by the enzyme phenylpyruvic decarboxylase to the flavorful compound phenylacetaldehyde. The main oxidation product of polyunsaturated fatty acids is 2,4-decadienal, which has a fatty taste. Thus, amino acids and ketoacids are the main precursors of protein-derived flavor substances [18]. The variety of aminotransferases and decarboxylases that are essential for the metabolism of amino acids may have originated from the microbial activities during dry curing. Thus, there may be a connection between microbial activity, endogenous enzyme activity, and flavor formation in dried salted fish.

### 2.2. Ketones

Ketones are produced through enzymatic degradation, the Maillard reaction, or microbial oxidation [19]. For example, 6-methyl-5-heptene-2-one is produced through the enzymatic oxidation of unsaturated fatty acids. 2-heptanone is produced through the oxidative degradation of polyunsaturated fatty acids and has a banana flavor. Ketones usually have a higher threshold, except for 6-methyl-5-heptene-2-one and 2, 3-octydione; 3, 5-octylodiene-2-one has an oily taste. In addition, ketone compounds can enhance the fishy taste of preserved fish. For example, 2-methyl-3-octanone only exists in raw meat and enhances the flavor of salted fish [20].

### 2.3. Unsaturated Alcohols

Unsaturated alcohols are produced through the autoxidation of unsaturated fatty acids. For example, the 13-hydroperoxide of linoleic acid is degraded to produce 1-octene-3-alcohol, with a mushroom flavor [21], and because its ROAV (Relative Odor Activity Value) is high under the five temperature conditions that can be detected, it is widely present in the volatile substances of fish and is a key volatile substance in dried and preserved fish products. 15-Lipid oxygenase oxidizes eicosapentaenoic acid to produce 1-pentene-3-ol, with a nutty, mushroom, or earthy flavor. 3-methylbutanol obtained through leucine decomposition has a greasy taste and can be used as a Maillard-reaction substrate. Unsaturated alcohols usually have a relatively low threshold and contribute to flavor formation in salted fish [22,23]. 1-heptanol has a nutty taste.

### 2.4. Esters

Esterification (the reaction of an acid with an alcohol) and alcoholysis (the reaction of an ester with an alcohol) are two ester synthesis pathways that are closely related to flavor [19]. The former is usually catalyzed by esterases (including esterase and lipase). The latter is catalyzed by acyltransferase or esterase. Esters can be obtained through the esterification of acids and alcohols. Ethyl acetate and isoamyl acetate are the characteristic flavor substances of salted and dried fish, and their thresholds are high, which can impart a fruit flavor to food, which gives salted fish a unique ester flavor [24]. Esterases play an important role. Esterase (EC 3.1.1.X) is defined as a class of enzymes that can catalyze the hydrolysis of water-soluble esters, including lipase (EC 3.1.1.1) and esterase (EC 3.1.1.3). Microbial esterase can release the free fatty acids and alcohols constituting esters while synthesizing ester compounds. The important flavor precursors of ester volatile compounds are free fatty acids and alcohols, which can also contribute directly to the overall aroma [25].

### 2.5. Other Flavor Substances

Although many kinds of hydrocarbon compounds are obtained through the alkoxy radical homolysis of fatty acids, they do not contribute much to flavor and can form aldehydes, ketones, and alcohols. Various alkanes (C5–C17) do not contribute much to the overall flavor because of their high threshold but are important intermediates of heterocyclic compounds that form the characteristic flavor of salted fish. Furan compounds, most of which have a meat flavor, have a very low threshold and are widely found in food. However, 2-ethylfuran and 2-amyl furan, which have strong milky and mung bean flavors, respectively, are found in small amounts in raw fish. In addition, amines are produced by salted fish products when the fish body rots and have an unpleasant odor. For example, trimethylamine usually causes the production of ammonia and a fishy taste, although low-salt lactic acid bacteria (LAB) can decompose or inhibit the formation of trimethylamine and other fishy flavor substances in the process of fermentation. Citric acid, sodium bicarbonate, and yeast are used to remove the fish taste in the process of salting [26].

## 3. The Role of Microorganisms

Microbial succession plays an important role in the formation of flavor during fermentation and is closely related to food safety. Currently, changes in the microbial community composition of salted and dried fish can be analyzed through amplicon sequencing and other methods. Changes in microbial communities have been attributed to changes in substrate composition and fermentation parameters, such as the salt concentration and fermentation temperature [27]. *Proteobacteria* and *Firmicutes* have been shown to dominate in salted and dried fish products [28]. Carbohydrate metabolism is reported to be closely related to the presence of *Firmicutes*, whereas amino acid and lipid metabolism is closely related to the presence of *Proteus*. Halophilic or halophobic microorganisms are more likely to exist in traditional fermented fish, such as halophilic o-omonas [29]. Cytoscape is used for bacterial correlation visualization, with more than four key flavor substances (ROAV ≥ 1), to identify key microorganisms by analyzing the correlation between bacteria and a volatile flavor via the Pearson correlation coefficient [30].

The complex microbial community plays an important role in flavor formation in traditional dry-cured fish. Based on the summary of the core microorganisms in the main salted and dried fish products (Figure 2), Halomonas and particularly *Staphylococcus* have been shown to be the most abundant genera detected in high-salt dried salted fish products. Specifically, to some extent, *Staphylococcus* species are the dominant microorganisms in these fish products. One study found *Staphylococcus,* with a relative abundance of 34.46%, to be the dominant bacterial genus [31].

*Staphylococcus* can affect the flavor and quality of dried salted and dried fish. When used as a starter culture, *Staphylococcus* promotes the rapid formation of flavor and delays the formation of fat oxidation [32]. Similar to LAB, *Staphylococcus* species have lipolytic and proteolytic activities and can slowly utilize carbohydrates and convert them into organic acids and aromatic substances, such as 2, 3-butanedione; acetaldehyde; and acetoin [33].

In the curing process of salted and dried fish, several metabolites produced by LAB and yeast are flavor substances. LAB can produce lactic, acetic, and propionic acids and can interact with alcohols and aldehydes in the fermentation process to form more complex flavor compounds and to promote the formation of flavor substances. LAB are used as starter cultures in most salted and dried fish products and degrade lipids and carbohydrates and act as biological preservatives [34]. A summary of the microbial community succession in salted and dried fish (Figure 2) shows that *Lactobacillus strains* (LAB) and *Saccharomyces* predominate. LAB includes *Lactobacillus*, *Lactococcus*, *Connostrea*, *Enterococcus*, *Pediococcus*, and *Lactococcus* among which *Lactobacillus* occupies a dominant position in the late curing process and is the dominant bacteria group. During salting, the organic acids (such as acetic and lactic acid) produced by LAB act as the main producers of acids and decrease the pH of the dried salted and dried fish system through glycolysis. Similar to *Staphylococcus* species, some LAB species have lipase and protease activities, contributing to the formation of dried salted and dried fish’s flavor compounds [35]. For example, *Lactococcus* lactis subsp. and Brevibacterium species have been shown to participate in aromatic amino acid metabolism [36]. In dry-cured products, *Lactococcus* and *Lactobacillus* produce a large amount of lipase, and the organic acids produced through metabolism promote the oxidation of fats into aldehydes, hydrocarbons, alcohols, and ketones, thus contributing to the flavor of the products [37].

Yeast can not only hydrolyze lipids but also can use fatty acids to synthesize esters and therefore has an important role in the formation of salted and dried fish’s flavor. Fungal communities are often dominated by *Saccharomyces cerevisiae*, *Kazachstania exigua*, *Torulaspora delbrueckii*, *Wickerhamomyces anomalus*, and *Pichia kudriavzevii*. Demaria and Candida albicans are common yeasts in fermented fish products. They can reproduce in an acidic environment and have a strong perfuming ability [38,39]. The metabolic activities of numerous bacteria are directly related to the flavor of dried salted fish, and the metabolic pathway of flavor compound breakdown is intimately tied to microbial enzymes. Lipid decomposition and oxidation and protein degradation are considered to be the key processes in the formation of dried salted fish’s flavor [40,41]. Through fat decomposition and oxidation, fats become flavor compounds or flavor precursors [42]. However, the excessive oxidation of fat can also lead to undesirable odors in dried fish products. The interaction of endogenous proteases and microbial enzymes cause protein degradation [43]. According to a study on lipid oxidation in black carp during storage, *Firmicutes*, *Proteobacteria*, and *Actinobacteria* were the most common bacterial phyla, with several being strongly linked to meat rotting [38].

During the process of protein degradation, for example, Belleggia et al. demonstrated that flesh-eating *Bacillus* can decarboxylate amino acids and degrade proline, leucine, and phenylalanine into methylpropanal, 3-methylbutanal, and phenylacetyl, respectively [44].

The KEGG (the Kyoto Encyclopedia of Genes and Genomes) database can also be used to investigate the relationship between microorganisms and flavor. Molecular technologies, such as proteomics, macrotranscriptomics, and metagenomes, are combined with a KEGG analysis to construct the relationship networks between microorganisms and flavor compounds. A genomic analysis has been combined with the KEGG metabolic pathway, metabolic pathways and microbial correlations of key enzymes, and microorganisms to analyze the protein-derived flavor substances and fat-derived flavor substances in dried salted and dried fish [45].

Salted and dried fish products are rich in protein, and protein degradation and amino acid metabolism are consequently important for flavor formation (Figure 2). Microbial metabolism can form flavor substances in products in two main ways. One is through transamination, whereby flavor precursors are formed from amino acids, such as aromatic amino acids (phenylalanine, tyrosine, and tryptophan), branched amino acids (leucine, isoleucine, and valine), and sulfur-containing amino acids (cysteine and methionine) [46]. Branched amino acids are first converted into alpha-ketoacids under the action of aminotransferases; these compounds are then converted into aldehydes through the catalytic action of ketoacid decarboxylate (KdcA) and are further converted into alcohols (Alchol) with the aid of dehydrogenase. Aldehydes can also be converted from hydroxyacid dehydrogenase to organic acids [47]. The second pathway is the elimination reaction, whereby methionine generates sulfur-containing compounds. The associated microorganisms involved in the reactions include *Bacillus* with amino acid transaminase, *Enterobacter* with branched-chain amino acid transaminase and with aromatic amino acid transaminase, *Macrococcus* with alpha-ketoacid decarboxylase, and *Lactococcus* with phenylpyruvate decarboxylase. Further studies have demonstrated that the protein flavoring substances are related to specific microorganisms during the processing of salted and dried fish.

Studies have been conducted on the protein-derived flavor substances in salted and dried fish (Table 1). Branched-chain amino acid metabolism includes leucine, isoleucine, and valine metabolism [48]. Leucine metabolism is mainly related to L-leucine aminotransferase (EC 2.6.1.6), and the related microorganisms are *Bacillus* and *Lactobacillus*. Isoleucine metabolism is related to L-amino acid oxidase (EC 1.4.3.2), although no related microorganism was found. Valine metabolism is mainly related to valine dehydrogenase (EC 1.4.1.23), and, again, no related microorganisms were found. In phenylalanine degradation, phenylalanine is first catalyzed by phenylalanine aminotransferase (EC 2.6.1.5) and aromatic amino acid aminotransferase (EC 2.6.1.57) and then undergoes an aminotransferase reaction to produce phenylpyruvic acid. Phenylalanine aminotransferase and aromatic amino acid aminotransferase are mainly associated with *Enterobacterium* and *Lactobacillus*. However, because of the relatively small abundance of *Lactobacillus*, the main microorganism activity is due to *Enterobacter* [49].

In salted and dried fish, the fat source is also one of the main sources of flavor (Figure 3). The initial reaction of flavor formation is the hydrolysis of fat into fatty acids by lipase and the release of free fatty acids (FFAs) for second-order fatty acid oxidation or reaction with proteins [50]. Lipoxygenases and lipoxygenases in adipose tissue or muscle fibers are the main endogenous lipases in adipose hydrolysis and oxidation. However, microorganisms have been shown to inhibit the activity of endogenous lipase through rapid fermentation to produce acid in the salting process. Therefore, in the process of fat oxidation, lipid lipoxygenase mainly oxidizes lipids, and automatic oxidation involving microorganisms also occurs. Although endogenous lipase is thought to be primarily responsible for lipolysis, the effect of microbial lipase on lipolysis cannot be ignored [51]. Microorganisms can also promote the release of FFA to a certain extent, and the relative action of endogenous and microbial enzymes can be studied by adding antibiotics to inhibit microbial growth [52].

Fat oxidation degrades unsaturated fatty acids into hydroperoxides (HPods), which can be further decomposed into low-molecular-flavor compounds. Moreover, HPods can react with amino acids or with Maillard-reaction intermediates to produce other odor substances, such as esters [53]. Arachidonic acid is metabolized by lipoxygenase (EC1.13.11) into 1-octylidene-3-alcohol, which can also be metabolized from linoleate to 13-HPOD by lipoxygenase (EC1.13.11) and then to hydroperoxide lyase (EC 4.1.2). In addition to the abovementioned linoleate ester that can generate 13-HPOD, this ester can generate 8-HPOD and 10-HPOD through lipoxygenase (EC 1.13.11). The 8-HPOD, 9-HPOD, 10-HPOD, and 11-HPOD produced via oleic acid metabolism by lipoxygenase (EC 1.13.11) are respectively metabolized by hydroperoxide lyase (EC 4.1.2) into capric aldehyde, heptal, nonal, and octyl, which are then, respectively, metabolized by alcohol dehydrogenase (EC 1.1.1.1) into n-decyl alcohol, heptal alcohol, nonyl alcohol, and octyl alcohol. Linolenic acid can be metabolized to 10-HPOD and 11-HPOD by lipoxygenase (EC 1.13.11) and to nonal and octyl compounds by hydroperoxide lyase (EC 4.1.2), respectively, and then by alcohol dehydrogenase (EC 1.1.1.1) to generate nonyl alcohol and octyl alcohol, respectively [48].

## 4. Fish Flavor Control and Humans: Future Directions

### 4.1. Application of New Flavor Detection Technology

Dried and salted fish is unique in flavor, with a diverse microbial community and complex metabolic pathways present during the curing process. Currently, most research has been conducted on the quality, processing, and flavor formation mechanisms of traditional dried salted fish. A flavor analysis database has been established based on the association of the flavor substances produced during different fermentation periods with the metabolic gene expression of the bacterial community using a combination of metagenomics, transcriptomics, and metabolomics to reveal the formation mechanism of the fermentation flavor. As the processing of dried salted fish can differ by region and as environmental factors have considerable influence on the curing process, it is necessary to use multiomics to comprehensively understand big data and to construct a fermentation flora prediction model to analyze and objectively restore the dried salted fish fermentation system. By clarifying the pattern of changes in bacterial flora and by regulating environmental factors, the fermentation process can be comprehensively and precisely regulated to understand the industrialization of dried salted fish products.

### 4.2. The Safety of Dry-Salted Fish Products

The function of the enzymes involved in fermentation should be investigated as well as the involvement of microbes in the fermentation process of dried salted fish. It is possible to isolate and purify microorganisms and the associated enzymes. Furthermore, fermenter strains with a great performance and the necessary enzymatic characteristics and kinetics can be chosen and studied further. Multiomics technologies like genomics and proteomics can be used together to create a microbial enzyme–nonvolatile metabolite–volatile flavor metabolic reaction network that can accurately regulate the taste and quality of dried salted fish products.

### 4.3. For Health

According to the different varieties of fresh fish raw materials, there can be different processing technologies, such as choosing the appropriate proportion of salt as well as the appropriate air drying time and strain. Due to the complexity of biological metabolic reactions, the health-related safety of the flavor quality of fish during production and storage can be further studied. Currently, improving the production process can meet the expectations of many consumers for healthy low-salt products. As consumers are paying more attention to health, low-salt dried fish has gained more focus in recent years. Salt plays an important role in the curing process; therefore, a series of measures are often required to minimize the flavor changes and quality deterioration caused by a low salt content. Consequently, extensive research has been conducted on new processes to improve the quality of products. The sensory quality of dried salted fish products can be improved through fermentation, which also has an inhibitory effect on harmful microorganisms during the curing process of low-salt dried fish. Artificially inoculated biological fermenters can be considered. For example, the addition of a single enzyme-producing microorganism or a certain percentage of complex microorganisms can accelerate the fermentation process of dried salted fish and improve its flavor, contributing to more standard products with better flavor. In addition, salt substitutes can be used; for instance, amino acids and salty peptides can be applied in low-salt technology. Thus, developing low-salt dried fish represents a current research hotspot with good application prospects.

In addition, the effects of the microorganisms in fish during storage could be evaluated by detecting the physical properties, microstructure, microbial properties, volatile flavor, and taste flavor of fish.

## Figures and Tables

**Figure 1 foods-12-03020-f001:**
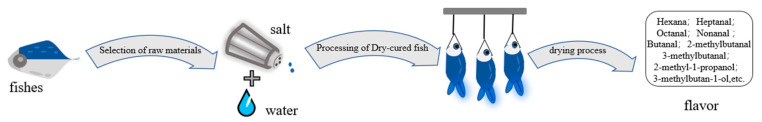
Process flow diagram of dry-cured fish.

**Figure 2 foods-12-03020-f002:**
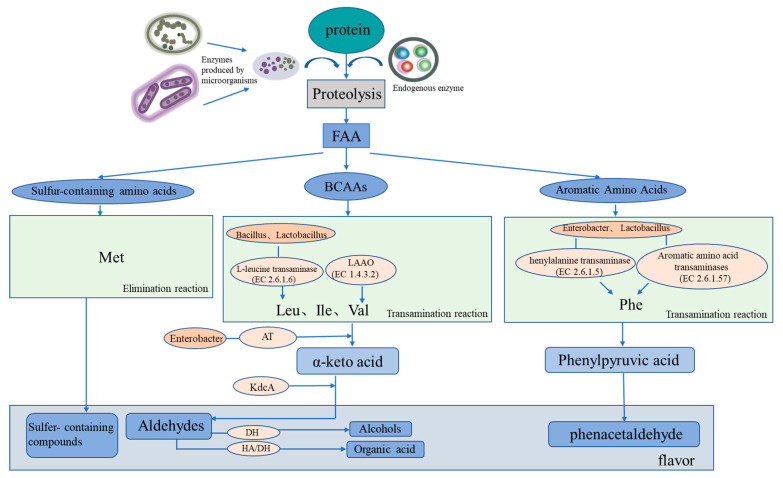
Mechanism of protein-derived flavor formation by microorganisms and endogenous enzymes in dry-cured fish. FAA represents free amino acids, BCAAs represents branched-chain amino acids, LAAO represents L-amino acid oxidase, and AT represents amine transaminase.

**Figure 3 foods-12-03020-f003:**
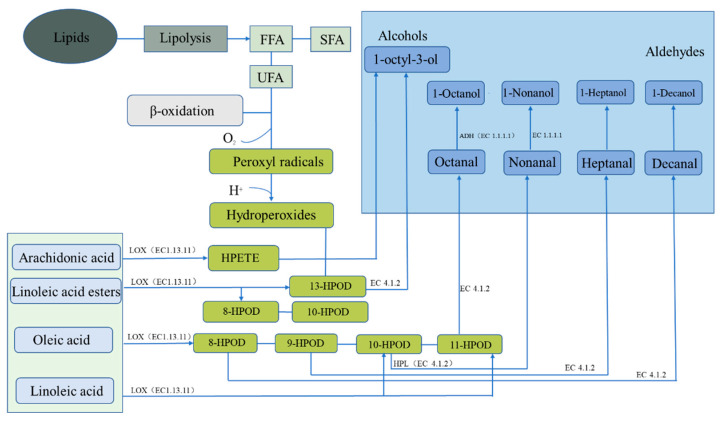
Fat metabolism pathway of endogenous enzymes forming flavor substances in dry-cured fish. ADH represents alcohol dehydrogenases, HPL represents hydroperoxide lyase, and LOX represents lipoxygenase.

**Table 1 foods-12-03020-t001:** Enzymes and microorganisms related to flavor of dry-cured fish.

Related Flavor Substances	Related FAA	Function	Enzyme	EC Number	Related Microorganisms
3-methyl-2-oxyvalerate acid	Ile	Degradation of BCAA; generation of α-Keto Acid	BCAT	2.6.1.42	*Enterobacteriaceae*; *Macrococcus*; *Lactobacillus*; *Staphylococcus* spp.
-	Val				
2-methylpropanol and2-methylpropanoic acid	L-val	Degradation of Val	VDH	1.4.1.23	-
3-methylbutanal, 3-methylbutanol, and 3-methylbutyric acid	L-leu	Degradation of Leucine; generation of α-ketoisocaproate	Leucine transaminase	2.6.1.6	*Bacillus*; *Lactobacillus*
		Degradation of Ile	L-amino acid oxidase	1.4.3.2	-
	Phe	Degradation of phenylpyruvate	Phenylpyruvate decarboxylase	4.1.1.43, 4.1.1.-	*Staphylococcus* spp.; *Lactobacillus*
phenylacetaldehyde	Phe	Degradation of phenylalanine; generation of phenylpyruvic acid	Aromatic amino acid transaminases II	2.6.1.57	*Enterobacteriaceae*; *Lactobacillus*
			Aromatic amino acid transaminases	2.6.1.58	
			Phenylalanine transaminase	2.6.1.5	-
			ALT	2.6.1.21	-
		Degradation of branched-chain amino acid source ketone acids	α- Keto decarboxylase	1.2.4.4	*Enterobacteriaceae*; *Staphylococcus* spp.

Note: BCAT represents branched-chain amino acid aminotransferases, and VDH represents valine dehydrogenases; alanine aminotransaminase.

## Data Availability

Data is contained within the article.

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
