# Peer review of "Flavor Formation in Dry-Cured Fish: Regulation by Microbial Communities and Endogenous Enzymes"

_foods, 2023, doi:10.3390/foods12163020_

Round 1

Reviewer 1 Report

Comments and Suggestions for Authors

Liu et al. present the flavor formation mechanism in dried salted fish. The article is interesting and provides a detailed overview of chemical changes in dry-cured fish. However, the sectioning is inadequate and misleading. Since this is the review, sections materials and methods, results, and discussion are confusing. The authors did not present original data. To make the article more reader-friendly the organization of the manuscript should be improved with clearer headings and subheadings, e.g., flavor substances in dried picked fish, and chemical changes occurring in fish, the role of microorganisms, the safety of dry-salted fish products. This would help guide the reader through the different sections. Authors should also emphasize the importance and novelty of their review.

Comments on the Quality of English Language

The English style is rather clear but some parts require corrections.

Author Response

      Thank you again for your positive comments and valuable suggestions to improve the quality of our manuscript. Thanks for your suggestion. We have tried our best to polish the language in the revised manuscript. The details are include in point by point.

Reviewer 2 Report

Comments and Suggestions for Authors

The manuscript “Flavor formation in dry-cured fish: Regulation by microbial communities and endogenous enzymes” has the structure of a research paper which is not the most adequate for a review. Thus, the structure of this manuscript must be completely revised.

Abstract

Page 1, lines 26-28 - This sentence seems too optimistic.

Page 1, lines 33-41 – This paragraph needs to be improved. Pickled fish is not the most correct designation for this type of products. Pickled fish refer to products prepared with vinegar.

Page 1, line 43 – It is scaling.

Page 1, line 44 – “Finished products” is not a “basic step”. Please check.

Page 2, lines 47-58 – Please revise this paragraph for its improvement.

Page 2, lines 67–69 – Please check this sentence because it is not clear.

Page 2, lines 89-90 – Figure 1 shows the names of a few aldehydes and ketones, but not the different types of products.

Page 3, lines 97 and 98 – Olefins basically design alkenes and pinene and d-limonene are included in terpenes.

Page 3, lines 98-103 - These sentences make reference to aldehydes which are further described in the next section so the text should be reorganized.

Page 3, lines 115-117 – Please revise this sentence for clarification.

Page 4, line 133 – Please clarify “is indicated” apart from the word “pickled”.

Page 4, line 146 – Thus ROAV stands for “Relative Odor Activity Value”? Please include the meaning of this acronym

Page 6, lines 235-258 - All these phrases repeat what has been said before.

Comments on the Quality of English Language

The English language need to be revised.

Author Response

(The authors gave the same response as above.)

Reviewer 3 Report

Comments and Suggestions for Authors

the article needs thorough re-structuring as it is a review article and has not used any statistical tools to devise results and indicate some sound conclusions on discussion of the results. 

The content is interesting and of value to the field of dried products but presentation is deceptive.

Comments on the Quality of English Language

The language can be improved as there are some very long sentences and by the end of the sentence the meaning completely changes. 

Some such sentences have been highlighted in the manuscript.

kindly rewrite the sentences and try to use simple short sentences with clear meaning. Confusing statements brings down the quality of content.

Author Response

(The authors gave the same response as above.)

Round 2

Reviewer 1 Report

Comments and Suggestions for Authors

The authors improved the paper significantly. It can be published after the last minor language editing.

Comments on the Quality of English Language

English language style has been significantly improved though there are some minor errors.

Author Response

We really appraciate you for your carefulness and conscientiousness. Your suggestions are really valuable and helpful for revising and improving our paper. We hope the correction will meet with approval. 

Reviewer 2 Report

Comments and Suggestions for Authors

The second version of the manuscript “Flavor formation in dry-cured fish: Regulation by microbial communities and endogenous enzymes” was improved but I still have the following suggestions of changes:

Page 1, line 44 – I suggest replacing the sentence “The packaging...” with “Finally, the dry-cured fish is packaged and ready for consumption”.

Page 2, line 52 – Please consider replacing the beginning of the sentence “To realize the…” with “To increase the production of dried fish and satisfy…”

Page 2, lines 55 and 56 – Please clarify the meaning of “high edible versatility” and “…is also very… meat products.”

Page 2, line 57 – I suggest deleting the word “substance”.

Page 2, line 60 – I suggest replacing “microbes” with “microorganisms”.

Page 3, lines 121 and 122 – Please consider this alternative: “Linolenic acid is converted into 11S-hydroperoxide by lipoxygenase, which…”

Page 4, line 138 – I think it is “dry-cured fish”.

Page 6, line 249 – Please check the sentence: “In addition, protein degradation”.

Page 9, line 355 – I suppose it is “metabolic reactions”. Please check.

Comments on the Quality of English Language

The English of this second version of the manuscipt was revised and improved. A few suggestions of changes are included.

Author Response

(The authors gave the same response as above.)
